# Gender Based Violence against Women in Sub-Saharan Africa: A Systematic Review and Meta-Analysis of Cross-Sectional Studies

**DOI:** 10.3390/ijerph17030903

**Published:** 2020-02-01

**Authors:** Muluken Dessalegn Muluneh, Virginia Stulz, Lyn Francis, Kingsley Agho

**Affiliations:** 1School of Nursing and Midwifery, Western Sydney University, Parramatta South Campus, Parramatta, NSW 2151, Australia; 2Amref Health Africa in Ethiopia, Addis Ababa 17022, Ethiopia; 3School of Nursing and Midwifery, Western Sydney University, Locked Bag 1797, Penrith, NSW 2751 Australia; 4School of Health Sciences, Western Sydney University, Locked Bag1797, Penrith, NSW 2571, Australia; 5African Vision Research Institute (AVRI), University of KwaZulu-Natal, Durban 4041, South Africa

**Keywords:** prevalence, GBV, IPV, non-IPV, physical violence, emotional violence, sexual violence, SSA, cross-sectional studies, and meta-analysis

## Abstract

This study aimed to systematically review studies that examined the prevalence of gender based violence (GBV) that included intimate partner violence (IPV) and non-IPV among women in sub-Saharan Africa (SSA). This evidence is an important aspect to work towards achieving the Sustainable Development Goals (SDG’s) target of eliminating all forms of violence in SSA. The Preferred Reporting Items for Systematic reviews and Meta-Analysis (PRISMA) guidelines were followed. Ovid Medline, CINAHL, Cochrane Central, Embase, Scopus and Web of Science were used to source articles with stringent eligibility criteria. Studies on GBV in SSA countries that were published in English from 2008 to 2019 were included. A random effect meta-analysis was used. Fifty-eight studies met the inclusion criteria. The pooled prevalence of IPV among women was 44%, the past year-pooled prevalence of IPV was 35.5% and non-IPV pooled prevalence was 14%. The highest prevalence rates of IPV that were reported included emotional (29.40%), physical (25.87%) and sexual (18.75%) violence. The sub-regional analysis found that women residing in Western (30%) and Eastern (25%) African regions experienced higher levels of emotional violence. Integrated mitigation measures to reduce GBV in SSA should focus mainly on IPV in order to achieve the SDG’s that will lead to sustainable changes in women’s health.

## 1. Introduction

According to the United Nations (UN), gender based violence (GBV) is defined as “any act of gender based violence that results in, or is likely to result in, physical, sexual, or mental harm or suffering to women, including threats of such acts, coercion or arbitrary deprivation of liberty, whether occurring in public or in private life [1].” GBV occurs and is classified in various ways. It can be defined depending on the relationship between the perpetrator and victim (intimate partner violence (IPV) and non-IPV), or by type of the act of GBV, such as sexual, physical or emotional violence [2]. This definition resonates throughout this manuscript.

GBV is a global public health problem that poses challenges in human health, with a higher prevalence in developing countries [3,4]. GBV not only plays a significant component in the morbidity and mortality of women, but this form of violence disproportionately affects the health status of women and their children [4]. GBV is an abuse of human rights that occurs internationally, in both developing and developed countries, regardless of culture, socio-economic class or religion [2,5,6] and varies in frequency, forms and extent from country to country [6].

It is often considered a ‘tip of the iceberg or silent epidemic’ as victims are hesitant to reveal their experiences of violence due to many barriers [7,8,9,10]. The barriers that women experience about reporting GBV include fear of stigma and shame, financial barriers, lack of awareness of available services, fear of revenge, lack of law enforcement action and attitudes surrounding violence as a normal component of life. Subsequently, this results in underreporting and challenges in accurately measuring the prevalence of GBV [7,10]. Overall, it is estimated that 30% of women have experienced at least one form of GBV in their lifetime since the age of 15 [4]. A World Health Organisation (WHO) multi-country study among women of reproductive age revealed that the overall prevalence of IPV ranged between 15% in urban areas (such as Japan) to 71% in provincial areas (such as Ethiopia) [3]. Evidence reveals that the problem is mostly prominent in developing countries where socioeconomic status is low and education is limited, especially in sub-Saharan Africa (SSA) countries [11,12].

The SDG’s are targeting eliminating all forms of violence against women and that all countries should be free from IPV by the year 2030, considering the deep rooted practices and effects of GBV against women [13]. In response to this, all stakeholders in all countries need to improve and work towards decreasing the prevalence of IPV [14]. Hence, better understanding of the prevalence of GBV is necessary for government and nongovernment organisations to inform an appropriate and effective policy response.

Despite the scope of this problem, most available studies are limited to developed countries with limited evidence focused on SSA countries [4,7,11]. Setting priority prevention and mitigation measures using the evidence from developed countries alone have substantial drawbacks [4,13]. In addition, studies conducted in SSA countries were focused on small-scale studies such as provinces and districts in particular countries that could overestimate the prevalence of GBV [3,4]. The small-scale studies conducted cannot be generalizable to the wider population. As a result, many SSA countries are yet to include the elimination of GBV on their policy agendas as a serious human rights violation with severe short and long-term implications [15]. There have been limited studies to date that have collectively and systematically examined the prevalence of GBV in varying forms among women aged 15-49 years of age in SSA countries, besides these small-scale studies.

Therefore, the aim of this research was to systematically determine the pooled prevalence rates of GBV including IPV and non-IPV in SSA countries. Additionally, the study analysed pooled prevalence rates of physical, sexual and emotional IPV in SSA countries. Findings reported in this study will provide vital evidence to inform policy and guide health investments to respond and prevent violence in alignment with the SDG’s target by 2030. In addition, the research findings will serve as a stimulus for further research on the dynamics of GBV in SSA countries to close existing gaps in the literature.

## 2. Materials and Methods

### 2.1. Study Setting

According to the United Nation (UN) World Population Review 2019, SSA consists of 48 countries with a population of 1,066,283,427 and accounts for 14.2% of the world population, with a growth rate of 2.66% in 2019 [16]. According to the UN sub-classification, regions are subdivided in to four regions including Western, Central African, Eastern, and Southern SSA [16]. Western SSA included Benin, Burkina Faso, Cape Verde, Gambia, Ghana, Guinea, Guinea-Bissau, Ivory Coast, Liberia, Mali, Mauritania, Niger, Nigeria, Senegal, Sierra Leone and Togo. Central African SSA included Cameroon, Central African Republic, Chad, Congo Republic-Brazzaville, Democratic Republic of Congo, Equatorial Guinea, Gabon, and Sao Tome and Principe. Southern SSA included Angola, Botswana, Lesotho, Mozambique, Namibia, South Africa, Swaziland, Zambia, and Zimbabwe [16]. The fourth least developed sub-region of SSA is Eastern SSA that included Burundi, Comoros, Djibouti, Eritrea, Ethiopia, Kenya, Madagascar, Malawi, Mauritius, Rwanda, Seychelles, Somalia, Somaliland, Tanzania and Uganda [16].

GBV is reported as a common practice in SSA and sexual violence prevalence is high in some countries such as Zambia (90%) and Ethiopia (711%) [3,17]. According to the Gender Equality Index Report, which includes data on reproductive health, employment, and empowerment, 27 of the 30 countries in the world that exhibit unequitable gender indices, are in Africa [13]. Most African cultural beliefs and traditions promote men’s hierarchical role in sexual relationships and especially in marriage [18]. Almost two-thirds (63%) of the African population live in remote rural settings that increases the difficulty to access basic amenities [16] and communities are disparate from the influence of central government or laws that prohibit GBV [13]. Only 22 African countries have adopted laws that prohibit GBV [14].

### 2.2. Information Source

A search of six electronic databases including Ovid Medline, CINAHL, Cochrane Central, EMBASE, Scopus, and Web of Science were undertaken. Relevant reference listings were checked, and grey literature was included, in addition to key research publications. Prior to starting this systematic review, the authors ensured the research question did not appear in any existing systematic reviews using Cochrane, Health Services Research Projects in Progress (HSRProj), and Prospero International Prospective Register of Systematic Reviews (PROSPERO) database registries.

### 2.3. Search Strategy

This systematic review was conducted using the Preferred Reporting Items for Systematic Reviews and Meta-Analyses (PRISMA) guidelines [19]. Pre-selected Medical Subject Headings (MeSH) terms and text words were used and searched in the above six databases for peer reviewed articles published between January 2008 and July 2019. The year 2008 was used as a baseline that provided increased global commitment of addressing GBV over the past decade [20]. There has been an increased uptake on the number of studies determining GBV prevalence internationally [20]. Moreover, the population dynamics have changed rapidly over the past ten years including improvements in health service access and education [21]. The search was limited to English language papers. Gender-based violence, intimate partner violence, domestic violence, spouse abuse, physical abuse, emotional violence, reproductive coercion, sexual assault, sub-Saharan countries, women aged 15-49 years, prevalence, magnitude and estimates were the key words used to conduct the search (Appendix A). The specified age of 15 years was used as a baseline as most studies used Demographic Health Surveys (DHS) that focused on women aged 15 to 49 years of age.

### 2.4. Eligibility Criteria

The following eligibility criteria were used to include studies in the systematic review:(i)Studies that reported the prevalence of GBV that focused on either or a combination of IPV, non-IPV, physical, sexual or emotional violence;(ii)Sample size greater than 300;(iii)Females within the age range of 15–49 years of age;(iv)Studies conducted in SSA countries [16] including countries in Western, Central, Eastern and Southern countries (see study setting for the list of countries);(v)Published in English from 2008 to 2019;(vi)Only quantitative studies.

### 2.5. Exclusion Criteria

GBV studies with no prevalence reported for example, studies that focused on factors associated with GBV; GBV consequences;Sample size less than 300;Qualitative studies not included as the main objective was to generate a pooled prevalence of GBV using the meta-analysis;Studies conducted outside SSA;Studies published before 1st of January 2008;Studies published other than English;Study participants less than 15 years of age or greater than 49 years of age.

### 2.6. Quality of Study

The quality of the studies that met the inclusion criteria was appraised using a Critical Appraisal Skills Programme (CASP) checklist for cross-sectional studies [22]. The following criteria were the key questions derived from the CASP to appraise the quality of the studies:Did the study address a clearly focused issue?Were the participants of the study recruited in an acceptable way?Was the outcome accurately measured to minimise bias?Was the sampling appropriate for the study?What are the results of the study?How precise are the tools used to measure the results?Do you believe the results?Can the results be applied to the local population?Are the results of the study relevant and fit with other available evidence?What are the implications of this study for practice?

The two independent reviewers rated the quality of each study by screening and considering the findings in relation to current practice or policy or relevant research-based literature and whether the findings can be transferred to other populations. The quality of each paper was rated using a ten-point scale using the CASP measurement criteria, 0 (none of the quality measures met) to ten (all quality measures met). The quality of the paper was based on the sum of points awarded. Studies were rated as poor quality (score ≤ 6); medium quality (7–8); and high quality (≥9) (See Appendix A.

### 2.7. Data Extraction

Endnote was used to manage search results. The authors reviewed the titles, abstracts, and keywords of every article retrieved by the search according to the selection criteria developed that included author, country, population/study subjects, study design, sample size and key findings and quality of the paper. The full texts of the articles were retrieved for further assessment if the information suggested that the study met the selection criteria or if there was any doubt regarding eligibility of the article based on the information in the title and abstract. Outcome data were extracted from studies using a tailored data extraction form adopted from various literature.

### 2.8. Data Analysis and Synthesis

This study was based on secondary data analysis. The syntax “metaprop” in Stata version 16.0 [23] was used to generate forest plots for each of the Appendix A. Each forest plot showed the prevalence of an indicator in individual authors and countries and its corresponding weight, as well as the pooled prevalence in each sub-region and its associated 95% confidence intervals (CI’s). A test of heterogeneity of the DHS and other data sets were obtained for the different authors and countries that showed a high level of inconsistency (I^2^ > 50%) thereby warranting the use of a random effect model in all the meta-analyses. Sensitivity analyses were conducted to examine the effect of outliers by using a method similar to that employed by Patsopoulos and colleagues [24] which involves comparing the pooled prevalence before and after elimination of one author or country at a time. Subgroup analysis was conducted by Eastern Africa, Western Africa and Southern Africa based on the UN classification [16]. The findings of the systematic review are synthesized and presented in summary form in Table 1.

### 2.9. Ethical Statement

This review used secondary data available in the public domain including the six electronic databases for the systematic review and the DHS dataset that are publicly available. Therefore, ethical approval was not required for this study because the data included in this analysis contained no identifying information and is publicly available and ethical approval has already been obtained by the original author or by the DHS program.

## 3. Results

A total of 4931 articles were found in the initial search from all databases. After removal of duplicates, 3275 remained for screening. Screening by title led to the exclusion of 3021 articles. Further reading of abstracts for 245 full-text articles led to the exclusion of another 187 articles. Twelve grey literature articles were included. Finally, 58 articles met the inclusion criteria (Appendix A).

### 3.1. Description of Included Studies

Fifty-eight articles were reviewed for data analysis and interpretation. The majority (95%) [4,9,15,21,25,26,27,28,29,30,31,32,33,34,35,36,37,38,39,40,41,42,43,44,45,46,47,48,49,50,51,52,53,54,55,56,57,58,59,60,61,62,63,64,65,66,67,68,69,70,71,72,73,74,75,76,77,78,79,80,81,82,83,84,85] of research articles included in this review were cross-sectional and the remaining (5%) were cohort studies [86,87,88]. Only cross-sectional studies were used to estimate the pooled prevalence rates. Overall, the total sample sizes ranged from 300 to 86,024 women of reproductive age (Table 1).

Overall, 58 cross-sectional studies investigated the prevalence of IPV either in the woman’s lifetime or over the previous year. Four studies reported non-IPV [4,27,60,80]. The studies that focused on IPV included 23 that reported physical violence, 18 that reported sexual violence and 20 studies that reported emotional violence. A relatively larger number of studies were found from Nigeria, South Africa, Kenya, Ethiopia and Uganda (Table 1). The assessment of the studies’ quality found that 30 (52%) were very good, 22 (38%) were medium quality and six (10%) studies were deemed low quality. The details of this assessment are provided as a Appendix A.

### 3.2. Prevalence of IPV among Women Aged 15–49 Years of Age

The prevalence of IPV in various SSA countries was sourced from 25 studies. The findings showed the prevalence ranged from as low as 13.9% (95% CI 10.8, 17.6%) [86] in a study conducted on perinatal women with depression symptoms in South Africa to as high as 97% (95% CI 94.6, 98%) [37] in a study conducted among rural women in Nigeria. The overall meta-analysis estimate for prevalence of IPV was 44.4% (95% 38.4, 49.8%) (Appendix A).

### 3.3. Prevalence of Intimate Partner Physical or Sexual Violence among Women Aged 15–49 Years of Age Using DHS Data (2008–2019)

Additional information was sourced from the most recent DHS reports that were conducted in SSA countries from 2008 to 2019 [101]. Only 29 sub-Saharan countries from the DHS reported on GBV. The prevalence focused on physical or sexual violence committed by a husband or partner against women [101]. We found that prevalence ranged from as low as 6.4 % in Comoros to 51% in Cameroun [101]. The meta-analysis showed a pooled prevalence of 31.3% (95% CI 26.3, 36.3) with heterogeneity detected among various surveys and countries (Appendix A).

### 3.4. Prevalence of Past Year IPV among Women Aged 15–49 Years of Age

A total of 18 studies investigated experiences of IPV over the past year among 24,941 women. The highest prevalence of IPV was found among women engaged in commercial sex work 78.7% (95% CI 75.2, 81.8%) in Kenya and in Nigeria (52.5%) (95% CI 46.7, 58.2%) [66]. Furthermore, a meta-analysis was estimated at 35.5 % (95% CI, 27.2, 44.12) (Appendix A). The sub-region analysis showed the highest pooled estimates in Eastern Africa (38.93%), followed by Western Africa (32%). Limited studies were sourced in South and Central SSA countries on IPV over the past year. Another sub-group analysis over the past year’s prevalence of GBV among pregnant and non-pregnant women showed the prevalence of experiencing any form of GBV amongst pregnant women was 30.5% (95% CI 21.2, 39.6) compared to non-pregnant women 39.8% (95% CI 26.98, 52.69) (Appendix A).

### 3.5. Prevalence of Physical IPV among Women Aged 15–49 Years of Age

Prevalence of physical violence was found in 23 studies ranging from 5.5% (95% CI 3.9, 7.9%) [47] among nurses in Ethiopia and 5.9% (95% CI 3.6–9.3%) [66] among HIV positive pregnant women in Nigeria to 59.9% (95% CI 55.5–63.8%) [29] among women in Uganda. A total of 66,361 women participants were included in the meta-analysis. Further, analysis of sub-regional estimates of physical violence was 29.3%, (95% CI 20.49, 38.09%) in Eastern Africa, 22.38% (95% CI 17.64, 27.12%) in Western Africa, 26.59% (95% CI 18.79, 34.38%) in Central Africa and 29.29% (95% CI 10.36, 48.18%) in Southern Africa. The overall pooled prevalence of physical violence was 26.14% (95% CI 21.69, 30.40%) with differences detected amongst the studies (Appendix A). Eleven studies showed that over the past year, the prevalence of IPV ranged from 9.3% (95% CI 8.3, 10.6%) in a Tanzanian study [96] to 43.8% (95% CI 39.5, 47.8%) [29] in Uganda. The pooled past year prevalence of physical IPV was estimated at 21.59% (95% CI 15.84, 27.33%).

### 3.6. Prevalence of Sexual IPV among Women Aged 15–49 Years of Age

Seventeen studies showed an overall prevalence of violent experiences and seven studies found experiences of sexual violence over the past year. Overall, pooled prevalence of sexual violence was 18.61% (95% CI 15.21, 22.00) with a high disparity among studies detected (Appendix A). The highest prevalence report was found in women in Northern Uganda (50%) (95% CI 46, 53%) [29], followed by a study conducted amongst women (39.7%) (95% CI 32.2, 47.2) in the Democratic Congo [31]. The lowest prevalence was found in Ghana (4%) (95% CI 3.1, 5.1) [37] and Nigeria (6.6%) (95% CI 6.3, 6.9) [45] amongst women of reproductive age. Similarly, a study conducted amongst nurses in Ethiopia showed one in 25 nurses (3.8%) had an experience of sexual violence (95% CI 2.5, 5.6) [47]. Eastern African women experienced relatively more sexual violence compared to other sub-regions. Among the seven studies with women experiencing sexual violence over the past year, violence ranged from the highest in Nigeria (42%) [66] and Ethiopia (31%) [39] to the lowest being 2% (95% CI 1.1–3.6 %) in a study conducted among HIV infected pregnant women in South Africa [93]. The results show there were no differences in lifetime and past year sexual IPV experiences (Appendix A).

### 3.7. Prevalence of Emotional IPV among Women Aged 15–49 Years of Age

There were 57,434 study participants included in the analysis. The prevalence of emotional violence was the highest among health care workers in Ethiopia (53.1%) (95% CI 48.7%, 57.4) [100] to Rwanda 9.7% (95% CI 8.8, 10.7) [56]. In particular regions, one in three women in most parts of Western Africa were emotionally abused by their partner. For instance, two studies conducted amongst women aged 15–49 years of age in Nigeria indicated the prevalence rate of emotional violence experienced was 44.4% (95% CI 40.9, 47.9) [58] and 34.7% (95% CI 29.5, 40.2) [28]. The most common type of violence was purported to be emotional violence in these countries in comparison to other regions. Sub-group analysis was conducted based on timing of the violence and found a pooled overall prevalence of emotional violence of 29.36% (95% CI 24.77, 33.9) and past one-year prevalence rate of 21.42% (95% CI 17.58, 25.26) (Appendix A). The test of heterogeneity and publication bias was detected (I^2^ = 98.9% and 88.6%, Egger’s test = 0.205).

Six studies have demonstrated the magnitude of emotional violence over the past year. The highest prevalence was found among female sex workers (31.9%) (95% CI 26.7, 37.1) [71] in Nigeria, followed by a study in Ghana (24.6%) (95% CI 20.5, 29.2) [98]. Correspondingly, a study conducted in Ethiopia showed one in five pregnant women experienced IPV over the past year [52].

### 3.8. Prevalence of Non-IPV among Women Aged 15–49 Years of Age

Non-IPV studies were rarely found. Of the total studies screened (58), only four studies investigated non-IPV. The highest non-IPV was found in Uganda (18.5%) and Somalia (16.5%) [60,80]. One out of six women reported experiencing physical and/or sexual violence by a non-intimate partner during their lifetime [60,80]. Two international studies showed that the prevalence of non-IPV was 11% (95% CI 4.5, 37.5) and 11.1 % (95% CI 8.5, 15.3) [4,5]. The pooled prevalence of non-IPV was 14.18 % (95% CI 11.61, 16.97) (Appendix A).

## 4. Discussion

This review incorporated all forms of GBV, including physical, sexual and emotional violence and IPV and non-IPV. The findings showed the pooled prevalence of GBV was high in SSA countries. This high pooled prevalence included almost half of the women experiencing IPV and a considerable number of females being abused by non-IPV. Emotional IPV violence was the most common type of violence in SSA. GBV was more prevalent in the sub-regions, in Western and Eastern Africa as compared to southern regions of SSA countries. Methodological quality of cross-sectional studies was appraised. We used only cross-sectional studies because we only found three cohort studies and/or randomized controlled trials.

Overall, a high pooled prevalence of IPV among women in SSA was found as compared to the global estimate which was conducted in 56 countries in 2013 [4] and SSA countries [5,21,102,103]. The findings of this review are comparable or slightly higher to studies conducted in 14 SSA countries [27,102]. The higher prevalence of IPV in our study could provide a better overview compared to previous studies where the number of countries involved were relatively small. Most importantly, this high prevalence might be due to the prevalence of gender inequality in regions for reasons including prerogative perceptions to males, tolerant attitudes in the community to IPV, poor education of women, female disempowerment and limited law enforcement in SSA [3,4,21,102,104,105].

Further analysis of the pooled prevalence rate over the past year revealed that more than two out of five women have reported experiencing IPV in SSA countries. This figure is consistent with a study conducted in other SSA countries [104] and more than five percent greater than the global lifetime prevalence of IPV (30%) [4]. This figure could be even higher, in reality, due to the underreporting associated with GBV [7] because of factors associated with fear of stigma, women preferring to keep quiet and fear of divorce, amongst many other reasons [6,7,14,17].

One of the interesting findings from this study is that the proportion (18%) of women affected during their lifetime and over the past year’s experiences of IPV were exactly the same as shown in Appendix A. This finding reflects that women in SSA countries are being subjected to experiences of violence continuously compared to other areas [14]. Overall, IPV in SSA countries is the most prevalent and challenging public health issue. The social context of the region is very complex and strong ties, extended family size and large communities of relatives are quite common that might expose women to potential perpetrators [106]. The prevention and management of GBV makes it more difficult in SSA countries.

The finding of pooled prevalence of IPV of the DHS was very high. There were statistical differences compared to the pooled prevalence of IPV computed from the electronic sourced articles. Moreover, the pooled prevalence from non-DHS studies found in electronic databases and DHS reviews were statistically different (*p* < 0.01). Our systematic review focused on IPV that included any of the combinations of physical, sexual or emotional violence or coexistence while DHS data focused on either physical or sexual violence among married women. In addition, DHS only explored married women, while in our study we used any population group in the age range of 15–49 years of age.

In this review, the pooled prevalence of all types of GBV, physical, sexual and emotional violence were consistently higher in SSA countries as compared to many other regions in the world [25,102,103]. Emotional violence was the most prevalent reported type of violence. Sexual experiences are reported not as frequently in many African countries for numerous reasons. The pattern of sexual violence is lower than emotional and physical violence, which might be related to victimized women being unlikely to report an attack due to fear of discrimination, feeling shame, and not being able to identify as well as physical violence [7,14].

One of the unexpected findings among health care providers was the highest prevalence rates (53%) of emotional violence and lowest prevalence rates (5%) of physical violence being reported in Ethiopia [100]. This high prevalence of emotional violence may be related to less satisfaction of service users due to long waiting times and less experienced health workers working in the health facilities. The majority of health care providers in the studies were females and this may be a reflection of gender inequality in the work areas. Most importantly, there is a lack of violence tracking or reporting mechanisms when it occurs among service providers, specifically focusing on emotional violence in the health care system [100,107]. Alternatively, the low prevalence of physical violence may be due to nurses having an understanding of the local context of GBV and being more likely to notify cases that would prevent perpetrators committing acts of violence [9,11]. Additionally, perpetrators may be unlikely to attack nurses at places such as a hospital or health centre where many other patients are receiving care from nurses.

The sub region analyses found that Eastern Africa (42%) including Ethiopia and Uganda were the most affected by all forms of IPV [29,47], followed by Western Africa (41.7%). In line with our findings, the two regions that experienced high prevalence rates of IPV in comparison to other African regions [7,25] was also consistent with other studies conducted in SSA countries [4,5]. In Eastern Africa, physical and sexual violence prevalence rates were worse and emotional violence prevalence rates were more common in Western Africa. This finding is consistent with findings of other studies [2,6,15,27]. This might be attributed to factors such as socioeconomic class, women’s disempowerment, community acceptance for wife beating and the type of community in which the study was conducted [4,6,27,29,108,109].

Alternatively, in Southern regions of Africa, the educational qualifications are relatively much better when compared to Eastern African countries [110]. A study conducted in South Africa found a combined intervention of economic intervention and education reduced IPV prevalence rates by 55% over a period of two years. Therefore, education differences could explain the differences of IPV prevalence in the two regions [110].

The pooled non-IPV prevalence (14%) experiences were very high. The pooled non-IPV prevalence experiences were slightly higher than the three studies that were conducted internationally, which was 11% [4,5,27]. The highest non-IPV prevalence may be related to political instability and war violence. For example, in Somalia the non-IPV prevalence was found to be 16.5% [80] which is mainly related to political instability and migration of the region. Moreover, some basic services are lacking, for example, health services, water and education. As a result, women are forced to travel long distances, which puts women more at risk to be subjected to violence as compared to those who have easy access and less travel time to those services.

### 4.1. Policy Implication

Findings reported in this study provide vital evidence to inform policy and guide health practitioners to respond and prevent violence in alignment with the SDG’s target by 2030. The aftermath of GBV has large ramifications for women’s health. It will be a challenge to achieve the SDG’s target to eradicate IPV by the year 2030, unless there is a timely intervention and policy designed for SSA regions. Governmental policies top priorities should focus on prevention of GBV, especially with the high prevalence of both IPV and non-IPV in all regions of SSA countries. This strategy needs to be supported by a legal framework to accommodate social support that includes educational and economic growth and provision of health information and services. All SSA countries need to develop an immediate action plan to support the challenges that women are facing with GBV. This review has added evidence to the current existing knowledge in the literature and has provided a stimulus for future research on the dynamics of GBV in SSA countries.

### 4.2. Strengths and Limitations of this Review

This is the first systematic review and meta-analysis to quantitatively summarize the prevalence of GBV that includes IPV and non-IPV that extends to SSA countries. A rigorous search was conducted from many electronic databases and selected nationally representative data sets (DHS) were used for most studies. A quality assessment was conducted with two independent reviewers conducting the quality screening. Only studies with adequate samples greater than 300 for representativeness were included in the review.

Despite the rigorous process of the systematic review, the searches only included articles published in English. The heterogeneity in our review could have been due to various factors such as different recall periods, underreporting, contextual differences including conflict, cultural differences and the quality of tools used to assess GBV. The generalizability of some small-scale studies is limited as studies may overestimate or underestimate GBV depending upon the context of the study. In addition, the number of studies on non-IPV were limited and it was difficult to identify the broader picture of GBV in the region. Furthermore, this review only included quantitative studies, most of which were cross-sectional. Therefore, qualitative studies were not included which may provide further information on the attitudes of women and communities about GBV that could indicate higher prevalence rates of GBV.

## 5. Conclusions and Recommendations

GBV against women is a pertinent health challenge in SSA countries. GBV that includes IPV and non-IPV are prevalent in SSA. More than two-fifths (44%) of women aged 15–49 years of age in SSA countries experienced some form of IPV and almost a fifth (14%) experienced non-IPV. All types of IPV (physical, sexual and emotional violence) are common experiences among women in SSA countries, with emotional violence being the most prevalent. Women living in Eastern and Western African regions experience the highest levels of GBV.

The need for an integrated mitigation measure to reduce GBV needs to be considered as a top priority in line with the SDG target in 2030 to reduce all forms of violence in SSA countries. Hence, government and private organisations should understand and address the problem of GBV. All organisations can allocate resources and design appropriate interventions that includes law enforcement to ensure social support is provided for women in the quest to eradicate GBV. In addition, more research is required to provide information on the dynamics of communities, the context, and associated factors of GBV and the subsequent effects of women’s reproductive health and beyond. Furthermore, more studies on IPV in SSA are required, especially in areas where political instability and war are on the increase.

## Figures and Tables

**Table 1 ijerph-17-00903-t001:** Characteristics of included studies (intimate partner violence (IPV) and non-IPV).

Authors	Country	Population	Sample Size	Study Design	Outcomes of the Results	Forms of GBV
Bleck et al. (2015) [25]	Selected SSA	Women aged 15–49 years	44,487	Cross-sectional	Approximately 29.0% (95% CI 28.8, 29.3) of women reported any physical or sexual IPV in their lifetime	IPV
Yaya et al. (2019) [26]	Angola	Women aged 15–49 years	7669	Cross-sectional	Overall, more than two-fifths of the women reported experiencing any IPV 41.1% (95% CI 38.7, 43.6): physical IPV 32.3% (95% CI 30.3, 34.5)) was most prevalent, followed by emotional 27.3% (95% CI 25.3, 29.4) and sexual IPV 7.4% (95% CI 6.6, 8.4)	IPV
Greene et al. (2017) [27]	14 countries in SSA	Women aged 15–49 years	86,024	Cross-sectional	Any form of lifetime IPV 42.5% (95% CI 32.5, 53.1), IPV was the most prevalent 36.5% (95% CI 26.5, 47.7); non-partner family violence 11.3% (95% 8.7,14.7) and non-family violence 3.2% (95% CI 2.3, 4.3); psychological IPV 25.1% (95% CI 19, 32.3) moderate physical violence 25.6% (95% CI 17.4, 36), severe physical IPV 8.9% (95% CI 5.8, 13.4), any sexual IPV 10% (95% CI 6.1,16.2).	IPV and non-IPV
Fawole et al. (2013) [28]	Nigeria	Women (street beggars and traders)	323	Comparative cross-sectional study	The lifetime experience of violence against women (VAW) was 66.3% (95% CI 62.5, 70.1) among the beggars and 54.8% (95% CI 52.2, 57.6) among the homemakers (*p* < 0.05). Psychological violence was experienced by 34.7% and 20.8% (*p* < 0.05); physical violence by 31.9% and 16.7% (*p* < 0.05) and sexual by 20.3% and 0.8% (*p* < 0.01) of the beggars and homemakers respectively.	IPV
Mootz et al. (2018) [29]	Uganda	Women aged 13 to 49	605	Cross-sectional	Both lifetime and previous year’s history: prevalence of experiencing IPV was psychological: 65.3% (95% CI 61%, 69) (life time) and 50.9% (95% 46.9, 54.9) (past 12 months); and physical: 59.9% (95% CI 55.7, 63-8) (lifetime) and 43.8% (95% CI 39.5, 47.8) (one year).	IPV
Vinck et al. (2014) [30]	Cote divore	Women aged 15 to 49 years	950	Cross-sectional	History of IPV 26.5% (95% CI 14, 36) reported experiencing IPV and 23.4% (95 %CI 16, 41) women reported past-year IPV.	IPV
Kirstenet al. (2010) [31]	DR Congo	Women (18–49)	998	Cross-sectional	Rates of reported sexual violence were 39.7% (95% CI 32.2, 47.2)	IPV
Ajah et al. (2014) [37]	Nigeria	Women aged 15 to 49 years	836	Cross-sectional study	The prevalence of domestic violence among rural women was significantly higher than that amongst urban women 97% (95% CI 94.6, 98) versus 81% (95 % CI 77, 84), (*p* < 0.001). In particular, the prevalence of physical violence was significantly higher among rural women than among urban women 37.2%, (95% CI 32.3, 42.4) versus 23.5 % (95 % CI 19.7, 27.6); (*p* < 0.05).	IPV
Adjah et al. (2016) [38]	Ghana	Women aged 15 to 49 years	1524	Cross-sectional	The proportion who ever had experienced domestic violence: 33.6% (95% CI 32, 36) emotional 30% (95% CI 27.7, 32.4); physical violence; 17% (95% CI 15.2, 19) and 4% (95% CI 3.1, 5.1) sexual violence	IPV
Admasu et al. (2016) [39]	Ethiopia	Women aged 15–49 years	300	Cross-sectional	IPV during recent pregnancy was 44.5% (95 % CI, 32.6, 56.4). About 55.5% (95% CI 157, 55.5) of women experienced all the three forms of intimate partner violence during recent pregnancy. Physical 29% (95 % CI, 24, 34.5), sexual 30% (95 % CI, 24.9, 35.6), and psychological 16% (95 % CI, 12, 20.7)	IPV
Shanko et al. (2013) [40]	Ethiopian	Women aged 15–49 years	858	Cross-sectional	Any experience of violence by an intimate partner was reported by 19.6% (95% CI 16.79, 22.2) and 70.3% of the perpetrators were husbands.	IPV
Fawole et al. (2018) [89]	Nigeria	Youth-students	640	Cross-sectional	At least one form of GBV was experienced: 86.7% (95% CI 83.9, 89.3) (89.1% of public and 84.8% private schools students (*p* = 0.32)). Psychological violence was the common type of GBV experienced (public—72.5% vs. private—69.2%; *p* = 0.37), while sexual violence was least (public—41.4% vs. private—37.4%; *p* = 0.3) prevalent.	IPV
Okenwa et al. (2009) [41]	Nigeria	Women aged 15–49 years	934	Cross-sectional	One-year prevalence of IPV was 29% (95% CI 26, 32), with significant proportions reporting psychological 23% (95% CI 20.4, 25.9), physical 9% (95% CI 7.3, 11), and sexual 8% (95% CI 6.4, 10.1) abuse.	IPV
Berhane et al. (2015) [90]	Ethiopia	Pregnant women	422	Cross-sectional	The prevalence of intimate partner physical violence in pregnancy was 20.6% (95% CI 16.70, 24.90).	IPV
Gust et al. (2017) [43]	Kenya	Women age greater than 18-49 years	7,421	Cross-sectional	Found that 11.8% (95% CI 11, 12.5) reported physical violence by a sexual partner in the last 12 months.	IPV
Kimani et al. (2016) [44]	Kenya	Women aged 15–19 years	301	Cross-sectional	Among the respondents, 33% (95% CI 27.6, 38.6) were victims of sexual violence.	IPV
Titilayo et al. (2017) [45]	Nigeria	Women aged 15–19 years	26,997	Cross-sectional	One-quarter (25%) (95% CI 24.5, 25.5) of the ever married women reported ever experiencing one form of domestic violence or the other (sexual 6.6% (95% CI 6.3, 6.9), physical 15.1% (95% CI 14.6, 15.5) and psychological/emotional 19.7% (95% CI 19.2, 20.2)	IPV
Pitipitan et al. 2013) [46]	South Africa	Women	1388	Cross-sectional	A total of 38.9% (95% CI 36.4, 41.5) reported a lifetime history of violence (i.e., ever being hit by a sexual partner). For recent violence, a total of 1140 (82.1%) reported not having been hit and a total of 17.9% (95% CI 15.9, 20.1) women did report having been hit by a sexual partner in the last four months.	IPV
Fute et al. (2015) [47]	Ethiopia	Nurses	660	Cross-sectional	Prevalence of workplace violence was 29.9% (95% CI 26.5, 33.5) of which physical violence accounted for 5.5% (95% CI 3.9, 7.6), verbal abuse for 26.4% (95% CI 23, 30) and sexual harassment for 3.8% (95% CI 2.5, 5.6).	IPV
Fesehan et al. (2012) [48]	Ethiopia	Women	422	Cross-sectional	The prevalence of physical violence in the last 12 months and lifetime was 25.5% (95% CI 21.3, 29.8) and 31.0% (95% CI 26.7, 35.7) respectively. The most common forms of physical violence reported included slapping 101 (61.6%) and throwing objects 32 (19.5%).	IPV
Fiorentino et al. (2019) [49]	Cameroon	Women (HIV positive women)	894	Cross-sectional	The prevalence of IPV was 29% (95% CI 26, 32) (emotional), 22% (95% CI 19.4, 24.9) (physical), 13% (extreme physical) and 18% (95% CI 15.6, 20.7) (sexual).	IPV
Bui et al. (2016) [91]	Zimbabwe		5280	Cross-sectional	Reporting physical violence: 27.11% (95% CI 25.9, 28), sexual 14% (95% CI 13.1, 14.97) and emotional 24.35% (95% CI 23.2, 25.5)	IPV
Pack et al. (2013) [92]	Kenya	Sex worker	619	Cross-sectional	About 78.7 % (95% CI 75.2, 81.8) of women reporting any IPV in the last 30 days.	IPV
Matsekeet al. (2017) [51]	South Africa	HIV positive women	673	Cross-sectional	Overall, 56.3% (95% CI 1 52.5, 60) reported having experienced either psychological or physical IPV, and 19.6% (95% CI 16.7, 22.8) reported physical IPV.	IPV
Azene et al. (2019) [52]	Ethiopia	Pregnant women	409	Cross-sectional	The prevalence of IPV during current pregnancy was found to be 41.1% (95% CI 36, 46). Of this, the prevalence of psychological, physical, and sexual violence was 29.1% (95% CI 27.1, 31.1), 21 % (95% CI 19.26, 22.9) and 19.8% (95% CI 18.0, 21.6) respectively.	IPV
Deyessa et al. (2009) [53]	Ethiopia	Women aged 15–49 years	1994	Cross-sectional	The lifetime prevalence of any form of IPV was 72% (95% CI 70.0, 73.9).	IPV
Gashaw et al. (2018) [54]	Ethiopia	Pregnant women	720	Cross-sectional	More than three quarters (80.6%) (95% CI 77.6, 80.5) reported to have a lifetime risk of emotional or physical abuse by their partner or someone important. The proportion of partner violence during the current pregnancy among ever exposed to IPV was 44% (95% CI 40.6, 47.4).	IPV
Berhanie [90]	Ethiopia	Pregnant women	954	Cross-sectional	About 40.8% (95% CI 37.6, 43.9) had experienced IPV during their pregnancy period. More than two thirds (68.6%) of cases had been exposed to IPV.	IPV
Berestein et al. (2016) [93]	South Africa	Pregnant women	632	Cross-sectional	Found that 21% (95% CI 18.4, 24.3) of women reported experiencing ≥1 act of IPV in the past 12 months, including emotional 15% (95% CI (12.4, 18.1), physical 15% (95% CI 12.4, 18.1) and sexual violence 2% (95% CI 1.1, 3.6). Of those reporting any IPV (*n* = 132), 48% reported experiencing two or more types. Emotional and physical violence was most prevalent among women aged 18–24 years, while sexual violence was most commonly reported among women aged 25–29 years	IPV
Pengpid et al. (2016) [55]	22 Countries (7-Africa)	Women aged 15–49 years	16,979	Cross-sectional	Cameroon 51.5%, Ivory Coast 30.2%, Madagascar 15.9%, Mauritius 6.7%, Namibia 21.5%, Nigeria 15.1%, South Africa 18.5%, Tunisia 6.4%	IPV
Finnoff et al. (2012) [56]	Rwanda	Women aged 15–49 years	4066	Cross-sectional	IPV: 37.1% (95% CI 35.6, 38.6), physical 33% (95% CI 31.5, 34.5), emotional 9.7% (95% CI 8.8, 10.7) and sexual 12.4% (95% CI 11.4, 13.5).	IPV
Sabri et al. (2019) [57]	Uganda	Women aged 15–49 years	7933	Cross-sectional	Recent IPV victimization was reported by 21.2% (95% CI 18.5, 20.1) of women.	IPV
Fawole et al. (2010) [58]	Nigeria	Ever married women	820	Cross-sectional	Lifetime prevalence of perpetration of physical abuse was 25.1% (95% CI 22.2, 28.3), while psychological violence was 44.4% (95% CI 40.9, 47.8). Two hundred and forty 29.3% (95% CI, 26.2, 32.5) had ever perpetrated sexual violence.	IPV
Agrdah et al. (2012) [59]	Uganda	Students	980	Cross-sectional	Overall: 10% (95% CI 8.2, 12.0) exposure to actual physical violence over the previous 12 months.	IPV
Perrin et al. (2012) [60]	Somalia	Women aged 15–49 years	2376	Cross-sectional	Among women, 35.6% (95% CI 33.4, 37.9) reported adult lifetime experiences of physical or sexual IPV and 16.5% (95% CI 15.1, 18.1) reported adult lifetime experience of physical or sexual non-partner violence.	IPV and Non- IPV
Chikhungu et al. (2019) [94]	Malawi	Women aged 15–49 years	24,562	Cross-sectional	Found that 42% (95% CI 41.4, 42.6) of ever-married women have experienced some form of violence perpetrated by their current or most recent spouse.	IPV
Wandera et al. (2015) [61]	Uganda	Women aged 15–49 years	1307	Cross-sectional	More than a quarter (27%) (95% CI 24.6, 29.3) of women who were in a union in Uganda reported sexual IPV.	IPV
Hatcher et al. (2019) [62]	South Africa	Women aged 15–49 years	2006	Cross-sectional	Currently partnered men, nearly half (48.4%) (95% CI 46.2, 50.6) perpetrated IPV.	IPV
Oumeora (2017) [63]	Nigeria	Women aged 15–49 years	500	Cross-sectional	Found that 13.6% (95% CI 10.8, 16.9) of the women had experienced domestic violence in the current pregnancy.	IPV
Schneider et al. (2010) [86]	South Africa	Women aged 15–49 years	425	Cohort	Found that 13.9% (95% CI 10.8, 17.6) reported IPV at baseline, with physical IPV being the most frequently reported (69.5%).	IPV
Tchokossa et al. (2018) [95]	Nigeria	Women aged 15–49 years	400	Cross-sectional	Findings showed that 55% (95% CI 49.9, 59.9) of the women have experienced at least one form of violence in their relationship but only 28% of the women who experienced IPV reported the act while 63.7% of those who did not report kept silent because they hoped their partner would change.	IPV
Stockl et al. (2010) [96]	Tanzania	Women aged 15–49 years	1503	Cross-sectional	Overall physical PV was 9.3% (95% CI 8.3, 10.6) during pregnancy. Seven (*n* = 88) and twelve per cent (*n* = 147) of ever-partnered, ever-pregnant women in Dares Salaam (*n* = 1298) and Mbeya (*n* = 1205), respectively, reported being physically assaulted during pregnancy by their partner.	IPV
Mahenge et al. (2016) [65]	Tanzania	Pregnant women	500	Cross-sectional	Overall 18.8% (95% CI 15.5, 22.6) experienced some physical and/or sexual violence during pregnancy. Forty-one women (9%) reported having experienced some physical and/or sexual violence at one to nine months postpartum.	IPV
Selin et al. (2019) [87]	South Africa	Adolescent girls and young women	2533	Cohort	The prevalence of IPV was nearly one quarter (19.5%) (95% CI 18.0, 21.2) of adolescent girls and young girls (AGYW) experienced any IPV ever (physical or sexual) by a partner. The prevalence of any IPV ever among AGYW aged 13 years to 14 years, 15 years to 16 years, and 17 years to 20 years was 10.8%, 17.7%, and 32.1%, respectively.	IPV
Ezeanochie, et al. (2010) [66]	Nigeria	HIV-seropositive pregnant women	305	Cross-sectional	The prevalence of IPV among the women was 32.5% (95% CI 27.5, 38.0), with psychological violence being the most common form of violence reported 27.5% (95% CI 22.7, 32.8) and physical violence the least reported 5.9% (95% CI 3.6, 9.3).	IPV
Prabhu, et al. (2011) [97]	Tanzania	women attending VCT	2436	Cross-sectional	Overall 17.7% (95% CI 16.2, 19.3) reported IPV during their lifetime.	IPV
Fawole et al. (2014) [71]	Nigeria	Female sex worker	305	Cross-sectional	The prevalence of VAW preceding the survey was 52.5% (95% CI 46.7, 58.2). Sexual violence was the most common type (41.9 %) (95% CI 36.4, 47.7)) of violence experienced, followed by physical violence (35.7%) (95% CI 30.4, 41.3) and psychological (31.9%) (95% CI 26.7, 37.1).	IPV
Addo et al.(2017) [98]	Ghana	Women aged 15–49 years	2000	Cross-sectional	About 34% (95% CI 29.3, 39.2) of respondents had experienced IPV in the past year, with 11.8% (95% CI 8.4, 16.4), 15.5% (95% CI 12.7, 18.7), and 24.6% (95% CI 0.5, 29.2) reported sexual, physical and emotional respectively. Past year experience of emotional and economic IPV were 24.6% and 7.4% respectively. Where lifetime experience was 50.9% (95% CI 46.0, 55.9), physical 32.2% (95% CI 28.3, 36.2), sexual 18.2 (95% CI 15.3, 22.7) and emotional IPV 34.5% (95% CI 29.7, 39.0).	IPV
Chen et al. (2017) [72]	Tanzania	Women aged 15–49 years	5371	Cross-sectional	In the past 12 months, there was 35% (95% CI 33.7, 36.3) that reported victimization among the study respondents.	IPV
Memiah et al. (2018) [99]	Kenya	Women aged 15–49 years	3028	Cross-sectional	Lifetime prevalence was 49.4% (95% CI 47.6, 51.1), (*p* < 0.001).	IPV
Schwitter et al. (2014) [73]	Uganda	Female sex workers	1467	Cross-sectional	Found that 82% (95 % CI 79, 84) experienced client-initiated GBV and 49% (95 % CI 47, 53) had been raped at least once in their lifetime. Physical violence 40% (95% CI 37, 43), verbal 45% (95% CI 42, 49), and sexual 50% (95% CI 46, 53).	IPV
Tusiime, et al. (2015) [74]	Uganda	Young pregnant women	416	Cross-sectional	Prevalence of sexual coercion was 24% (95 % 20.0, 28.6) and was higher among those who had non-consensual sexual first time experiences (29.0%) compared with those who had consensual sexual first time experiences (22.6%).	IPV
Onoh, et al. (2013) [75]	Nigeria	Pregnant women	321	Cross-sectional	Found that, 44.6% (95% CI 39.5, 50.6) reported having been abused in pregnancy.	IPV
Falb, et al. (2014) [88]	Côte d’Ivoir	Women aged 15–49 years	981	Cohort	Half (49.8%) (95% CI 46.6, 53.1) of all women reported lifetime physical or sexual IPV, and nearly 1 in 5 (18.6%) reported experiencing reproductive coercion.	IPV
Mutagom et al. (2019) [80]	Rwanda	Female sex workers	1978	Cross-sectional	A high proportion of female sex workers (FSW’s) were physically abused multiple times 42.6% (95% CI 40.2, 44.8). During sex work, 35.6% faced physical violence, and 14.8% faced this physical violence many times. Physical violence happened in the last month preceding the survey in 25.4% FSW’s; it occurred in the last 12 months in 49.7% FSWs. When asked about the last time FSWs faced physical violence, most (63.1%) of the perpetrators were clients; however, in 12.5% of (95% CI 11, 14) cases, the perpetrator was a member of law enforcement. A large proportion 18.3% (95% CI 14.6, 17.9) had been sexually abused outside of the family circle.	IPV and non- IPV
Bamiwoy et al. (2014) [81]	Multicounty	Women aged 15–49 years	38,426	Cross-sectional	The overall prevalence of any form of violence (physical, sexual or emotional) ranged from 30.5% in Nigeria to 43.4% in Zimbabwe; 45.3% in Kenya; 45.5% in Mozambique; 53.9% in Zambia and 57.6% in Cameroon	IPV
Yenealem et al. (2019) [100]	Ethiopia	Healthcare workers	531	Cross-sectional	The prevalence of workplace violence was found to be 58.2% (95% CI, 53.7, 62.3) in which verbal abuse 53.1% (95% CI 48.7, 57.4) followed by physical attacks 22.0% (95% CI 18.6, 25.6) and 7.2% (95% CI 5.1, 9.8) sexual harassment. Females are most exposed in all forms of workplace violence: verbal abuse 161 (57.1%), physical attack 69 (59.0%) and sexual harassment 38 (100%) when compared with men.	IPV
Hendricks et al. (2018) [85]	Tanzania	Female sex workers	496	Cross-sectional	Overall 40% (95% CI 35.6, 44.4) of participants experienced recent physical or sexual violence, and 30% recently experienced severe physical or sexual violence.	IPV
Garcia- et al. (2013) [4]	Global (56 countries)	Women aged 15–49 years	11,594	Cross-sectional	Lifetime prevalence of physical and/or sexual IPV among ever-partnered women by WHO region (African region)—36.6(95% CI 32.7, 40.5); lifetime prevalence of non-partner sexual violence by WHO region—8.5% (95% CI 15.3%, 45.6) proportion of women reporting IPV and/or non-partner sexual violence.	IPV and non- IPV

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
