# Peer review of "Gender Based Violence against Women in Sub-Saharan Africa: A Systematic Review and Meta-Analysis of Cross-Sectional Studies"

_ijerph, 2020, doi:10.3390/ijerph17030903_

Round 1

Reviewer 1 Report

The article is consistent and responds to the knowledge needs of the updated world. The question arises: why did the authors exclude the review of qualitative design articles? I believe that this aspect is not argued and could present partial information.

Author Response

The question arises: why did the authors exclude the review of qualitative design articles? I believe that this aspect is not argued and could present partial information.

Thanks for the feedback; we revised the objective section as: Therefore, the aim of this research was to systematically determine the pooled prevalence rates of GBV including IPV and non-IPV in SSA countries. Additionally, the study analysed pooled prevalence rates of physical, sexual and emotional IPV in SSA. Findings reported in this study will provide vital evidence to inform policy and guide health investments to respond and prevent violence in alignment with the SDG’s target by 2030. In addition, the research findings will serve as a stimulus for further research on the dynamics of GBV in SSA countries to close existing gaps in the literature.

Reviewer 2 Report

The study is about a very interesting topic in an understudied context. The manuscript is a systematic review that examine the prevalence of Gender-Based Violence (GBV) in 16 Sub Saharan Africa (SSA). Authors have followed the PRISMA procedure and did a search on six electronic databases including Ovid Medline, CINAHL, Cochrane Central, 80 EMBASE, Scopus, and Web of Science. The main strength of this paper is that it highlights a topic of major social relevance in a little-studied context.

However, the paper has important limitations that should be addressed. The aim of the study is not well justified. 

The Authors should emphasize and better organize the criteria that they have used for the systematic review. 

The authors should give more contextual information about SSA countries and GBA.

The authors should provide more information about the items of the ten-point scale used to measure the relevance of each paper. 

English and writing should be revised. For instance:

This review aimed to systematically review. Web of Science (WOS) instead of Web Science. Check subheadings number

Author Response

Review 2: Comments and Suggestions for Authors

The study is about a very interesting topic in an understudied context. The manuscript is a systematic review that examine the prevalence of Gender-Based Violence (GBV) in Sub Saharan Africa (SSA). Authors have followed the PRISMA procedure and did a search on six electronic databases including Ovid Medline, CINAHL, Cochrane Central, EMBASE, Scopus, and Web of Science. The main strength of this paper is that it highlights a topic of major social relevance in a little-studied context.

1)       The aim of the study is not well justified. 

Thanks for the feedback; we revised the objective section as: Therefore, the aim of this research was to systematically determine the pooled prevalence rates of GBV including IPV and non-IPV in SSA countries. Additionally, the study analysed pooled prevalence rates of physical, sexual and emotional IPV in SSA. Findings reported in this study will provide vital evidence to inform policy and guide health investments to respond and prevent violence in alignment with the SDG’s target by 2030. In addition, the research findings will serve as a stimulus for further research on the dynamics of GBV in SSA countries to close existing gaps in the literature.

a.       The Authors should emphasize and better organize the criteria that they have used for the systematic review.

Thank you. We have revised the inclusion and exclusion criteria based on your feedback. We have revised this in the method section.

2)       The authors should give more contextual information about SSA countries and GBA.

Thank you. We inserted a section in the method section under 2.1. Study Setting of the manuscript to provide the contextual information in the revised version.

3)       The authors should provide more information about the items of the ten-point scale used to measure the relevance of each paper. 

Thanks for the feedback. We have included a table to show this 10-point information in the Supplementary material Table S2.  We also included in the method section about the key CASP questions.

4)       English and writing should be revised. For instance:  This review aimed to systematically review. Web of Science (WOS) instead of Web Science. Check subheadings number 

We have revised the grammar and others  

Reviewer 3 Report

The article provides significant contribution on prevalence of GBV in parts of SSA. The text is clear and understandable enough. Before publication, the following should be considered:

1) discrepancy in reporting used keywords (see page 1 and first paragraph of page 3);

2) some syntax flaw through the text; also redundancy in some places;

3) review eligibility and exclusion criteria: a) the Democratic Republic of Congo missing from point iii, b) exclusion of studies lacking “adequate sample” -see subtitle 4.1;

4) the author reports GBV that might be driven by political and war issues in Somalia, this has been the case for the DRC as well and there’s a strong body of literature available on GBV in that country -it once was referred to as the world capital of rape; further exploration of the state of GBV in that country would have shown more works on non-IPV;

5) Based on the limited number of articles focusing on non-IPV included in this analysis, the author should be cautious on the conclusion drawn on point 4 (discussion)

6) the author should be cautious with the language style used in subtitle 4.1, second last sentence as one may find it to be exaggerating and patronizing;

Author Response

Comments and Suggestions for Authors

The article provides significant contribution on prevalence of GBV in parts of SSA. The text is clear and understandable enough. Before publication, the following should be considered:

Discrepancy in reporting used keywords (see page 1 and first paragraph of page 3); Key words (Page 1 and Page 3):

Thank you for the feedback. We included the most pertinent key terms due to the guidelines of the journal recommending ten words to provide an overview of the contents of the article. The search terms were not all the same words that were included in the key terms as they were the specific words used in the search, for example, some of the key words included cross-sectional and meta-analysis which described the type and analysis of the manuscript, but not used in the search terms to investigate the type of articles in which we were interested to analyse.

Some syntax flaw through the text; also redundancy in some places;

Thank you. We have revised the manuscript using your suggestions.

Review eligibility and exclusion criteria: a) the Democratic Republic of Congo missing from point iii, b) exclusion of studies lacking “adequate sample” -see subtitle 4.1;

Thanks for this feedback. We have now separated the eligibility and the exclusion criteria and the sample size has now been included. The list of countries now includes the Democratic Republic of Congo that has now been moved to the Study Setting section under point 2.1.

the author reports GBV that might be driven by political and war issues in Somalia, this has been the case for the DRC as well and there’s a strong body of literature available on GBV in that country -it once was referred to as the world capital of rape; further exploration of the state of GBV in that country would have shown more works on non-IPV;

Thank you for feedback. Yes, we agree that there is a strong body of literature available on GBV in the DRC, and we have reported that sexual violence (IPV) was 39% in Table 1 in the DRC which was quite significant. However, the DRC analysed studies separately in relation to effects of conflict related violence and other forms of violence, so we were unable to establish any inferences in the area of non-IPV. We have included this in the conclusion and future recommendations of the manuscript, last sentence: Furthermore, more studies are needed in SSA on non-IPV especially in areas where the political instability and war are high.

Based on the limited number of articles focusing on non-IPV included in this analysis, the author should be cautious on the conclusion drawn on point 4 (discussion)

Thank you. It is well noted and changed in the body of the manuscript.

the author should be cautious with the language style used in subtitle 4.1, second last sentence as one may find it to be exaggerating and patronizing;

Thank you. It is well noted and changed in the body of the manuscript.

Round 2

Reviewer 2 Report

The second version has improved. I think it can be accepted. 

Reviewer 3 Report

Please accept the revised version of the article.